# COVID-19 vaccination uptake among healthcare workers in Ghana: A comprehensive analysis of knowledge, attitude, perceived vaccine effectiveness, and health belief model constructs

**Whiteson Mbele**[1,2]◎*, **Phyllis Dako-Gyeke**[1]◎, **Andreas Ndapewa Frans**[1]◎

1 Department of Social and Behavioral Sciences, School of Public Health, College of Health Sciences, University of Ghana, Accra, Ghana, 2 Kasiya Mission Hospital, Pemba District Health Office, Pemba, Southern Province, Zambia

◎ These authors contributed equally to this work.

* whitesonmbele@gmail.com

**Data Availability Statement:** All relevant data are within this paper and publicly available at doi:10.5061/dryad.vdncjsz3d.

## Abstract

The novel Coronavirus Disease 19 (COVID-19) caused devastating effects globally, and healthcare workers were among the most affected by the pandemic. Despite healthcare workers being prioritized in COVID-19 vaccination globally and in Ghana, hesitancy to receive the vaccines resulted in delayed control of the pandemic. In Ghana, healthcare workers had a vaccine acceptance of 39.3% before the vaccine rollout. Consequently, this study assessed the uptake of COVID-19 vaccination and associated factors among healthcare workers in Ghana in the post-vaccine roll-out period. This was an analytical cross-sectional study that used a semi-structured questionnaire to collect data on COVID-19 vaccination uptake and influencing factors. 256 healthcare workers were selected in Ayawaso West Municipality of Ghana using a stratified random sampling approach. Descriptive statistics were used to examine socio-demographic factors and Likert scale responses. Bivariable and Multivariable logistic regression were performed using IBM SPSS version 22 to identify predictors of vaccine uptake and a statistical significance was declared at p<0.05. More than three-fourths of participants 220 (85.9%) had received at least one dose of the COVID-19 vaccination, while 36 (14.9%) were hesitant. More than half 139 (54.3%) had adequate knowledge about COVID-19 vaccination and the majority 188 (73.4%) had positive perceptions about its effectiveness. Moreover, 218 (85.2%) of HCWs had a positive attitude towards COVID-19 vaccination. Positive attitude towards COVID-19 vaccination (AOR = 4.3; 95% CI: 1.4, 13.0) and high cues to action (AOR = 5.7; 95% CI: 2.2, 14.8) were the factors that significantly predicted uptake of COVID-19 vaccination among healthcare workers. COVID-19 vaccination among HCWs in Ghana is promising. However, hesitancy to receive the vaccination among a significant proportion of HCWs raises concerns. To ensure vaccination of all healthcare workers, interventions to promote vaccination should target key determinants of vaccination uptake, such as attitude towards the vaccination and cues to action.

**Funding:** The study was funded by WHO/TDR as part of the Postgraduate Training Programme 2022/2023. The funders played a role in determining the study topic and design. However, the funders had no role in data collection and analysis, decision to publish, or preparation of the manuscript.

**Competing interests:** The authors have declared that no competing interests exist.

## Introduction

Coronavirus disease 19 (COVID-19) is caused by severe acute respiratory syndrome Corona virus 2 (SARS-CoV-2) [1]. COVID-19 was first detected in China in the city of Wuhan in 2019 and the disease was declared a pandemic by the World Health Organization (WHO) in March 2020 [2]. This was the first documented pandemic caused by a Coronavirus in history [3]. Since the identification of the outbreak, over 774 million cases have been reported worldwide with over 7 million deaths as of January 2024 [4]. COVID-19 has caused many social and economic disruptions, and the impact was felt across all countries worldwide. To prevent infection and minimize the impact of COVID-19, countries across the world developed emergency response mechanisms such as social distancing, face masking, shutdown of public facilities, ban of public gatherings, lockdowns and COVID-19 vaccination [5]. The implementation of these measures varied in different countries and was mainly determined by country-specific burden of the disease and the availability of resources. In Ghana, immediate measures were instituted to contain and prevent the spread of the disease, and these included a ban on all public gatherings, mandatory wearing of face masks, the closure of schools and religious places of worship, partial lockdown, and education campaigns on COVID-19 preventive measures [6]. These measures were critical in mitigating the impact of the disease [5]. The burden of COVID-19 was much higher in developing than in developed countries [7]. Ghana recorded its first case of COVID-19 in March 2020, and by June 2020 over 17,763 cases were confirmed [6, 8]. Healthcare workers (HCWs) were unavoidably exposed to the virus due to direct contact with patients [9]. This places them at increased risk of infection and mortality. Several studies have indicated high infection rate and mortality among HCWs [10–14]. In a systematic review and meta-analysis, the prevalence of COVID-19 infection among HCWs globally was estimated to be 11% [10], a cause for global health concern. The impact of the pandemic was higher in Africa, with a prevalence of infection among HCWs of up to 45% in some countries [15]. Ghana is one of the countries in Africa that reported high COVID-19. In a case-control study involving 2402 HCWs in Ghana, nurses comprised the majority of cases, accounting for 61.0% of those infected, followed by medical doctors (11%) [16]. However, the odds of contracting COVID-19 infection were higher among doctors compared to nurses. Another study conducted in Ghana revealed a significant burden of the pandemic among medical doctors, with a prevalence of COVID-19 infection among this group estimated at 8.9% and a case fatality rate (CFR) of 1.7 deaths per 100 infections [17]. The CFR reported in Ghana is higher than that observed in Africa (CFR = 1.2) and globally (CFR = 0.9) [18].

Infection of HCWs led to shortage of workforce due to isolation, quarantine, hospitalization, and death of frontline HCWs and this caused a significant human resource shortage, placing an additional burden on already struggling healthcare systems [19, 20]. COVID-19 vaccines have been highly effective in reducing mortality and morbidity in countries with high vaccine acceptability rates [21, 22] (Alhassan, Aberese-Ako, et al., 2021; Solante et al., 2023). As of January 2024, over 13.5 billion COVID-19 vaccines have been administered globally [23]. WHO developed guidelines to COVID-19 vaccination and recommended that health workers should be prioritized for vaccination due to their high risk of infection. In response, several countries prioritized healthcare workers in vaccination [24]. Despite COVID-19 vaccine availability, there has been a global hesitancy to receive the vaccine among the general population, with vaccine acceptability as low as 23% in some countries [25]. Vaccine hesitancy refers to delay in accepting the vaccine or refusing safe vaccines despite its availability [26]. COVID-19 vaccine hesitancy is prevalent even among healthcare workers, with a global vaccine acceptance ranging from 27.7% to 77.3% [27]. In Africa, vaccine acceptance has been worryingly low, with some countries reporting acceptance as low as 6.9% [28]. As of February

2024, only 32.7% of the population in low and middle-income countries had received at least one dose of the vaccine [29]. This is despite the availability of COVID-19 vaccines in developing countries through initiatives such as the Vaccine Global Access [30].

Globally, healthcare workers have generally adequate knowledge and a positive attitude towards COVID-19 vaccination [31–35]. Despite the possession of good knowledge and a positive attitude about COVID-19 vaccination among HCWs, the challenge of vaccine hesitancy is still prevalent, with only 48% of HCWs in Africa reported to have accepted the vaccine [36]. Targeted vaccination of healthcare workers who are at increased risk of acquiring infection is an effective infection control measure. During the rollout of the COVID-19 vaccines, countries across the world prioritized healthcare workers in receiving the vaccines [24]. However, vaccine hesitancy among healthcare workers across the world stunted the uptake of this intervention. Globally, vaccine acceptance among healthcare workers ranged from 23% to 97% [25]. In Ghana, studies conducted before rolling out the COVID-19 vaccines revealed differing vaccine acceptance rates among HCWs. One study documented an acceptance rate of 39.3% [37], which was lower than the 46.0% acceptance rate reported across the African region [36]. Conversely, two other studies found acceptance rates of 70.0% [22] and 78.6% among HCWs in Ghana, exceeding the regional acceptance rate in Africa [38]. Key determinants of vaccine hesitancy among HCWs, particularly in developing countries included concerns about vaccine effectiveness [39], knowledge about the vaccine [40, 41], attitude towards vaccination [35, 42, 43], and health belief model measures, including perceived susceptibility to and severity of COVID-19 infection [44, 45], perceived barriers and benefits to vaccination [42], and cues to action [46]. Globally, perceived COVID-19 vaccine effectiveness among the general population ranged from 67.8% to 95.9%, and factors such as age, sex, level of education, and marital status significantly influenced individual perception of vaccine efficacy [47]. While studies conducted before the roll-out of the COVID-19 vaccination in Ghana indicated mixed results on COVID-19 vaccine acceptance rate among HCWs as indicated earlier, no studies were conducted to ascertain the actual uptake of COVID-19 vaccines among HCWs in Ghana after the vaccines were rolled out. This limitation hinders the ability to develop targeted policies aimed at enhancing future vaccination rates within this crucial group. HCWs are the main sources of health information to communities and role models of health-related behaviors; hence their hesitancy to receive vaccinations has a negative influence on the general population. Consequently, this study assessed the uptake of COVID-19 vaccination and influencing factors among HCWs in Ayawaso West District, Ghana after the vaccines were rolled out. Ayawaso West District was the epicenter of the pandemic with the highest recorded number of COVID-19 cases (63.9%) in Ghana during the first wave of the pandemic [8]. Results from this study can aid in developing effective health promotion interventions to boost COVID-19 vaccination uptake among HCWs in Ghana and beyond.

## Conceptual framework

Behavior change theories and models are important tools that have been applied in public health to understand factors that influence health-related behavior [48]. The Health Belief Model (HBM) has widely been used successfully in understanding the vaccination behavior of populations against Influenza and COVID-19 viruses respectively [42, 46, 49]. This model states that health-related behavior is dependent on several factors, namely, perceived susceptibility, perceived severity, perceived benefits, perceived barriers, cues to action, and self-efficacy [50].

Perceived susceptibility refers to the extent that an individual believes they are at risk of acquiring the disease. Individuals who believe they are vulnerable to getting infected with

COVID-19 have perceived susceptibility. Perceived severity refers to individual beliefs about the consequences of getting the disease. Individuals who feel threatened with the consequences of getting infected with COVID-19 have a high perception of risk. With regards to COVID-19 vaccination, Perceived benefit refers to the belief that receiving the vaccine will reduce the risk of getting infected with COVID-19 or the seriousness of the disease threat. Perceived barriers to COVID-19 vaccination refer to individual beliefs about restrictions to vaccination, and these are related to psychological, physical, or financial factors. Cues to action, with regards to COVID-19 vaccination refer to factors that are considered necessary or triggers to receiving the vaccine [51].

The HBM has been useful in explaining and predicting behavior related to COVID-19 vaccination acceptance. Several studies globally have found significant associations between HBM constructs and acceptance of COVID-19 vaccination [42]. Perceived susceptibility, perceived severity, perceived benefits, and cues to action have positively been associated with vaccine acceptance, while perceived barriers have inversely been associated with vaccine acceptance. In this current study, the HBM provided the theoretical framework that was used to assess determinants of COVID-19 vaccination uptake among HCWs (Fig 1). Five Constructs of the HBM (Perceived susceptibility, perceived severity, perceived benefits, perceived barriers, and cues to action) guided the development of the questionnaire. Socio-demographic characteristics such as age, profession, marital status, previous experience with COVID-19 infection, and knowledge about COVID-19 vaccination can influence an individual's perception of susceptibility to and severity of COVID-19 infection. Awareness of COVID-19 vaccination and its benefits may influence adoption of the vaccination.

## Materials and methods

### Study design, setting, and population

This study was conducted in Ayawaso West municipal district, which is one of the 29 administrative districts in the greater Accra region of Ghana. The district has a population of 75,303 based on the 2021 population and housing census, of which 38,164 are males and 36,689 are females. The most reported diseases in the area are malaria, typhoid, and diarrhea [52]. The first index case of COVID-19 in Ghana was reported in this district [8]. Currently, the district has 2 public hospitals; Legon hospital and the University of Ghana Medical Center (UGMC), and 5 community health planning and services (CHPS) zones located in the five sub-municipalities, namely, Abelemkpe, Dzorwulu, Legon, Roman Ridge and Westland. This cross-sectional study involved healthcare workers from Legon Hospital, UGMC, and from the 5 CHPS zones. Healthcare workers comprised workers who provided direct and indirect care to clients and included medical doctors, nurses, midwives, pharmacists, laboratory technologists, radiographers, dentists, dieticians, optometrists, and administration staff.

### Inclusion and exclusion criteria

The inclusion criteria were healthcare workers aged 18 years or above from the two public hospitals (UGMC and Legon Hospital) and from the 5 CHPS zones in Ayawaso West district. These included medical doctors, nurses, midwives, administration staff, and allied health workers (i.e. dental therapists, dieticians, opticians, pharmacists, radiographers, laboratory technologists, and physiotherapists. Healthcare workers who met the inclusion criteria but were on leave from work and not present during the time of data collection, sick and unable to participate in the study, or disagreed to participate in the study were excluded.

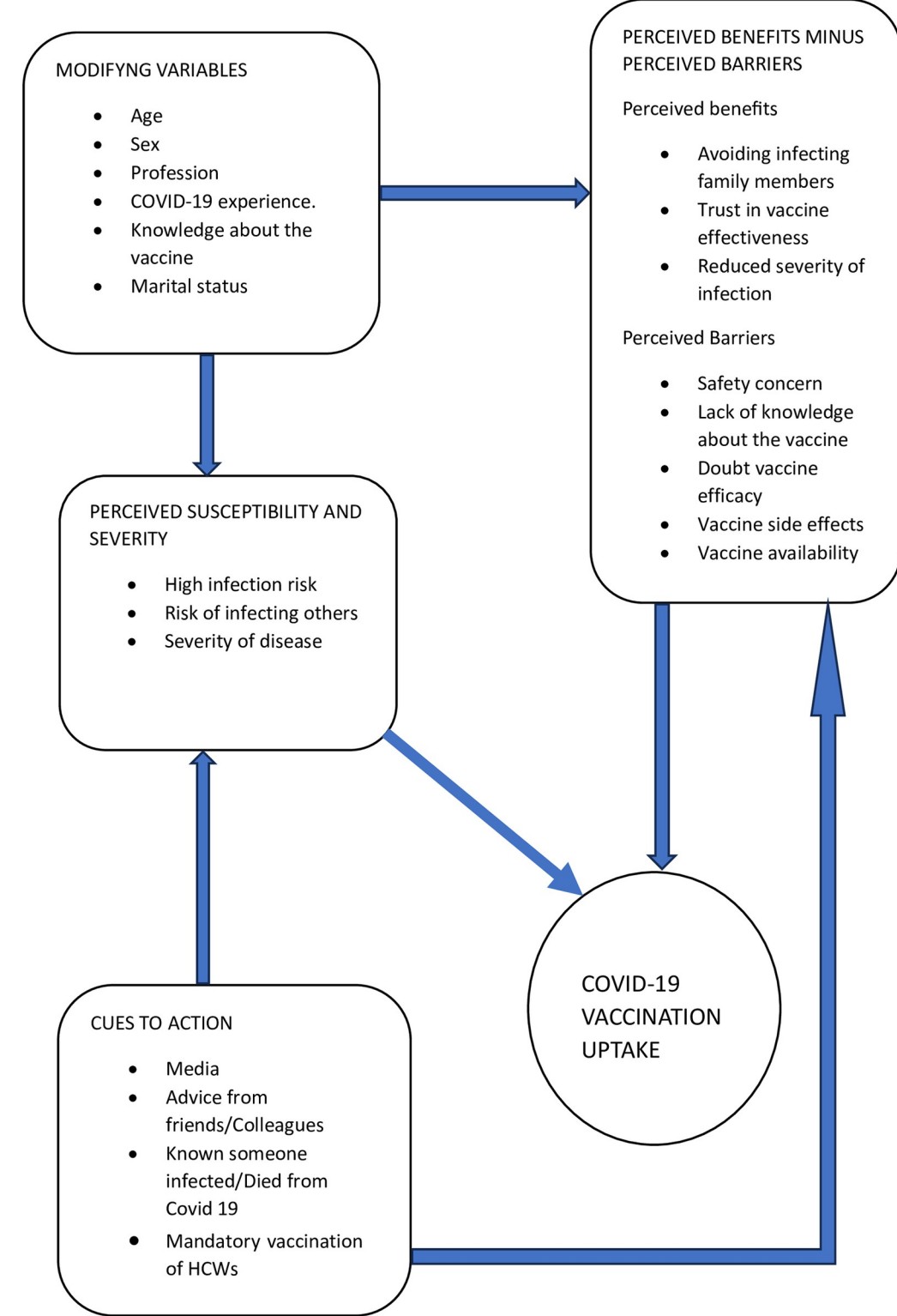

**Fig 1. Conceptual framework.**

## Sample size calculation and sampling method

The sample size was calculated using the Yamane formula [53]. A confidence level of 95% and a level of precision of 5% were assumed in this study. The formula is as follows.

$$n = N/[1 + N(e)^2]$$

Where n is the desired sample size, N is the estimated population of healthcare workers in Ayawaso West Municipality, and e is the level of precision. Ayawaso West Municipality is in the Greater Accra region, which had an estimated 20,344 healthcare workers as of 2018 [54]. The estimated total number of healthcare workers in Ayawaso West municipality was 702. Using the formula, the sample size was calculated as follows.

$$n = 702/[1 + 701(0.05)^2]$$

$$n = 255$$

To account for the possibility of non-response and incomplete questionnaires, an additional 10% was added to the sample size [55], giving a sample size of 255 + 10% of 255 = 280.5 ~ 281.

Proportionate Stratified random sampling was used to select study participants. This provided a more representative sample of the study population. Initially, lists of healthcare workers were obtained from the human resource offices at each health facility, identifying 843 eligible participants. The sample size for each health facility was then determined using proportional allocation based on the number of facility health workers, with UGMC having 405 eligible participants, Legon Hospital having 396, and CHPS having 42. Consequently, 135 participants were allocated to UGMC, 132 to Legon Hospital, and 14 to CHPS zones. Subsequently, the workforce was stratified based on professional designation, resulting in five distinct strata: nurses, midwives, medical doctors, administrative staff, and allied health workers. The latter category included professionals such as physiotherapists, radiographers, dieticians, opticians, pharmacists, laboratory technologists, and dental therapists. The sample size to be recruited from each profession type was then proportionally allocated. Thereafter we used the sample () function in R to randomly select participants from each stratum of healthcare workers. Selected participants were contacted and those who consented to participate in the study were issued with a questionnaire.

## Data collection tool and quality control

Data was collected using a self-administered questionnaire adapted from previously published studies on COVID-19 [56–59]. The tool was designed in English, the official working language of the study setting, and was divided into six sections. Section 1 assessed socio-demographic characteristics, section 2 assessed knowledge about COVID-19 vaccination, section 3 assessed attitude towards COVID-19 vaccination, section 4 assessed perceived effectiveness of COVID-19 vaccines, and Section 5 assessed five domains of the health belief model (HBM), including perceived susceptibility, perceived severity, perceived benefits, perceived barriers, and cues to action. The adapted questionnaire was reviewed for face and content validity by an expert from the University of Ghana, School of Public Health. The resultant questionnaire was pilot-tested among 20 HCWs at Legon Hospital to verify its simplicity and feasibility. Modifications were made to the questionnaire based on the pre-test results. Those who participated in the pilot test were excluded from the final analysis.

We assessed the internal consistency of the adapted questionnaire using Cronbach's alpha, and a measure of $\geq 0.7$ was considered a satisfactory measure of internal consistency [60].

Knowledge about COVID-19 vaccination was assessed by asking 10 questions related to vaccination and each question had answer options of true, false and I don't know. The correct option was assigned a score of 1 and all other options were assigned a score of 0. The score for this section therefore ranged from 0 to 10. To categorize participants as having good knowledge and poor knowledge, bloom's cut-off point of ≥80% (8 or more correct responses) was used and participants who scored ≥80% were considered to have good knowledge and otherwise have poor knowledge [61]. Attitude toward the COVID-19 vaccination was assessed with six attitude-based questions, while perceived vaccine effectiveness was assessed based on three questions. For the HBM, five dimensions were assessed, which included perceived susceptibility (five questions), perceived severity and seriousness (five questions), perceived benefits of the vaccine (seven questions), perceived barriers to vaccination (twelve questions), and cues to action (eight questions). Items on Attitude towards COVID-19 vaccination, perceived vaccination effectiveness, and HBM constructs were measured on a five-point Likert scale, in which a score ranging from 1 to 5 was given from "strongly disagree" to "strongly agree" (1 = strongly disagree, 2 = disagree, 3 = neutral, 4 = agree, 5 = strongly agree). To minimize acquiescence bias, some questions were reverse asked. These were reverse-coded during the analysis. The total score for each dimension was computed and the mean score for each domain was calculated. Scores above the mean indicated greater levels of the specific domain and otherwise lower levels [44].

Data was collected from 24th July 2023 to 31st August 2023 with the help of three trained assistants, who are healthcare workers. One research assistant was recruited from each of the two public hospitals (UGMC and Legon Hospital) and the third research assistant was recruited from the CHPS zone to assist with the recruitment of study participants, distribution, and collection of completed questionnaires from the two hospitals and at the CHPS zones respectively. The research assistants were trained on the objectives of the study, contents of the questionnaire, and ethical considerations through a one-day workshop.

## Outcome measure

The outcome variable was COVID-19 vaccine uptake, which was defined as having received at least one dose of any of the current available COVID-19 vaccines. This was assessed by asking whether the participants received the vaccine or not with 'yes, one dose', 'yes, two doses', and 'No' response options. A binary variable was created and those who received at least one dose of the vaccine were categorized as having accepted the vaccination while those who had not received any dose of the vaccine were considered hesitant.

## Data management and analysis

Completed questionnaires were coded and data was entered into Microsoft Excel after a manual check for completeness. The entered data was imported into IBM SPSS version 22 for analysis. The study population was characterized using descriptive statistical analyses (mean, standard deviation, and frequency). Bivariate analysis was performed between socio-demographic characteristics and the outcome variable (uptake of COVID-19 vaccination) using the Chi-square test to determine associations between socio-demographic factors and uptake of COVID-19 vaccination and a P-value of <0.05 was considered statistically significant. Univariable logistic regression analysis was performed to identify variables to include in multivariable logistic regression with a statistical significance set at p<0.2. A cut-off p-value of <0.2 ensured that our multivariable regression analysis included all potentially important predictive variables [62]. In multivariable logistic regression analysis, a statistically significant association was considered at p<0.05. The model was assessed for goodness of fit using the Hosmer and

Lemeshow test, and we ensured that multicollinearity was not problematic by checking the variance inflation factor (VIF), with a threshold of <5 [63]. Crude odds ratios (COR) and adjusted odds ratios (AOR) with a 95% confidence interval (95% CI) are reported.

## Ethical considerations

This study was approved by the Ghana Health Service Ethics Review Committee (GHS-ERC) with protocol ID: GHS-ERC 052/05/23 and by the University of Ghana Medical Center Institutional Review Board (UGMC-IRB) with protocol ID: UGMC-IRB/MSRC/050/2023. Additionally, clearance to collect data from health facilities was obtained from Ayawaso West Municipal Health Directorate and from managements of selected hospitals. Written and informed consent was obtained from participants prior to collecting data. The collected data was anonymous, and the information obtained in this study was kept confidential. To safeguard the confidentiality of acquired information, completed questionnaires were enclosed in sealed envelopes before collection by data collectors. No name or any identifying information was asked or taken during the data collection process. Physical copies of data, such as questionnaires and signed consent forms, were stored in a locked cabinet to prevent unauthorized access and all electronic data were stored on a password-protected computer accessible only to investigators. Participants were free to withdraw from the study at any point and only those participants who consented were recruited in the study. COVID-19 protocol was adhered to during the data collection process.

## Results

### Socio-demographic characteristics of participants

Table 1 provides descriptive characteristics of the participants. To have a more representative sample, the study involved diverse categories of health workers (i.e., Nurses, Doctors, Midwives, Hospital administrators, and Allied health workers including physiotherapists, dentists, pharmacists, radiographers, opticians among others) and from different levels of care (i.e., Hospitals and CHPS Zones). We recruited participants from two public hospitals (UGMC, and Legon Hospital) and from five CHPS zones. Almost equal numbers of participants were recruited from UGMC 123 (48.0%) and Legon Hospital 120 (46.9%) while 13 (5.1%) were recruited from CHPS zones. Overall, 281 healthcare workers completed the survey, of which 256 (91.1%) were included in the final analysis (the rest were excluded due to unanswered or incompletely answered questionnaires). The mean age of participants was 31 years (SD = 6.2), and the majority were females 163 (63.7%).

About one-third 77 (30.1%) of the participants were nurses and allied health workers made up a quarter 65 (25.3%) of the participants. A large proportion of participants 160 (62.5%) were not married and more than half 146 (57%) were frontline healthcare workers. Close to half 121 (47.3%) of the participants had 1–4 years of working experience. Only a small proportion of 96 (37.5%) of participants had previously tested positive for COVID-19. We have recorded a high rate of 220 (85.9%) of COVID-19 vaccination among healthcare workers in Ghana (Table 1).

### Knowledge about COVID-19 vaccination

Table 2 shows the responses of participants to questions related to knowledge about COVID-19 vaccination. The majority of participants 234 (91.4%) were aware of the vulnerability of elderly people and those with chronic diseases to COVID-19 infection and hence the need for them to get vaccinated. More than three-quarters 223 (87.1%) of participants knew that

**Table 1. Socio-demographic characteristics of healthcare workers in Ayawaso West Municipality (n = 256).**

| Variables | Frequency (N) | Percentage (%) |
|---|---|---|
| **Age group** | | |
| Less than 30 years | 101 | 39.5 |
| 30 to 35 years | 89 | 34.8 |
| Above 35 years | 66 | 25.7 |
| **Sex** | | |
| Male | 93 | 36.3 |
| Female | 163 | 63.7 |
| **Profession type** | | |
| Nurses | 77 | 30.1 |
| Midwives | 45 | 17.6 |
| Medical doctors | 23 | 9 |
| Allied health workers[b] | 65 | 25.3 |
| Administration staff | 46 | 18 |
| **Years of practice** | | |
| <5 years | 121 | 47.3 |
| 5–10 years | 79 | 30.9 |
| >10 years | 56 | 21.8 |
| **Marital status** | | |
| Married | 96 | 37.5 |
| Not married | 160 | 62.5 |
| **Frontline health worker** | | |
| Yes | 145 | 56.6 |
| No | 111 | 43.4 |
| **Previous diagnosis of COVID-19** | | |
| Yes | 96 | 37.5 |
| No | 160 | 62.5 |
| **COVID-19 vaccination status** | | |
| Vaccinated | 220 | 85.9 |
| Not vaccinated | 36 | 14.1 |
| **Health Facility** | | |
| UGMC | 123 | 48.0 |
| Legon hospital | 120 | 46.9 |
| CHPS | 13 | 5.1 |

[b] Physiotherapists, radiographers, dieticians, opticians, pharmacists, laboratory technologists, dental therapists

vaccines work by allowing the immune system to build a memory against infectious agents and close to two-thirds 161 (62.9%) were aware that RNA and DNA vaccines give genetic codes that enable the immune system to produce a memory against infectious agents. Additionally, the majority 208 (81.3%) of participants were aware that people with chronic diseases such as diabetes, hypertension, and heart diseases are eligible to receive the COVID-19 vaccines. Participants had low levels of awareness 100 (39.1%) on whether pregnant and lactating mothers are eligible to receive the COVID-19 vaccines or not. The mean knowledge score for this scale was 7.4 (SD = 1.8). More than half 139 (54.3%) of participants had good knowledge about COVID-19 vaccination at a cut-off of 80% of the total score of 10, whereas 117 (45.7%) had poor knowledge. The Cronbach alpha ($\alpha$) for this scale was 0.52.

**Table 2. Knowledge about COVID-19 vaccination among HCWs in Ayawaso West Municipality of Greater Accra, Ghana, 2023 (n = 256).**

| Question | Correct Response | Incorrect Response | Do not know. |
|---|---|---|---|
| | n (%) | n (%) | n (%) |
| Vaccines are effective in combating highly contagious diseases[t] | 212 (82.8) | 14 (5.5) | 30 (11.7) |
| Traditionally, vaccines create immunity by introducing a weak form of an infectious agent that allows the immune system to build a memory against this agent[t] | 223 (87.1) | 10 (3.9) | 23 (9.0) |
| The RNA and DNA vaccines give our bodies the genetic code it needs to allow our immune system to produce the antigen on its own[t] | 161 (62.9) | 23 (9.0) | 72 (28.1) |
| Covid-19 vaccines are being developed as quickly as possible, but they were required to receive the necessary regulatory licenses[t] | 172 (67.2) | 16 (6.3) | 68 (26.6) |
| The flu vaccine protects against covid-19[f] | 136 (53.1) | 32 (12.5) | 88 (34.4) |
| People with chronic diseases and elderly are more likely to have the disease and its complications, so they should get the vaccine[t] | 234 (91.4) | 11 (4.3) | 11 (4.3) |
| Young people are healthy and therefore do not need to follow preventive measures and to get the vaccine in order to protect themselves against Covid-19[f] | 235 (91.8) | 17 (6.6) | 4 (1.6) |
| Patients with chronic diseases like diabetes, hypertension and heart diseases are eligible to take the COVID-19 Vaccine[t] | 208 (81.3) | 15 (5.9) | 33 (12.9) |
| Pregnant and lactating mothers are eligible to take the COVID-19 vaccine[t] | 100 (39.1) | 94 (36.7) | 62 (24.2) |
| Until the readiness and the availability of COVID-19 vaccine, we cannot do anything to tackle the disease[f] | 212 (82.8) | 29 (11.3) | 15 (5.9) |

[t] True statement

[f] False statement

## Attitude towards COVID-19 vaccination

Table 3 shows responses of participants to questions related to attitude towards COVID-19 vaccination. The mean attitude score was 20.9 (SD = 4.6). The Cronbach alpha for this scale was 0.84. Of the total participants, 218 (85.2%) had positive attitude towards COVID-19 vaccination whereas 38 (14.8%) had negative attitude.

## Perceived COVID-19 vaccine effectiveness

Table 4 shows the responses of participants to questions related to the perception of the effectiveness of COVID-19 vaccination. The mean perceived COVID-19 vaccine effectiveness score was 9.7 (SD = 3.1). The Cronbach alpha for this scale was 0.74. More than two-thirds of

**Table 3. Attitude towards COVID-19 vaccination among HCWs in Ayawaso West Municipality of Greater Accra, Ghana, 2023 (n = 256).**

| Question | Strongly Disagree n (%) | Disagree n (%) | Neutral n (%) | Agree n (%) | Strongly Agree n (%) |
|---|---|---|---|---|---|
| Do you think that many diseases prevented by vaccination are serious ones, mainly infectious? | 9 (3.5) | 30 (11.7) | 35 (13.7) | 123 (48.0) | 59 (23.0) |
| Do you think that the immunity acquired after contracting the disease is better than after vaccination? * | 25 (9.8) | 87 (34.0) | 95 (37.1) | 42 (16.4) | 7 (2.7) |
| Do you think that it is better to wait for the next emerging vaccines than to get one of those developed in the first stage? * | 37 (14.5) | 94 (36.7) | 87 (34.0) | 28 (10.9) | 10 (3.9) |
| Would you make a decision not to vaccinate for reasons other than illness or allergy? * | 49 (19.1) | 108 (42.2) | 55 (21.5) | 32 (12.5) | 12 (4.7) |
| Would you delay getting vaccinated for reasons other than illness or allergy? * | 48 (18.8) | 113 (44.1) | 39 (15.2) | 45 (17.6) | 11 (4.3) |
| Do you think that opinions on vaccines are primarily governed by the opinions and benefits of pharmaceutical companies? * | 30 (11.7) | 80 (31.1) | 93 (36.3) | 42 (16.4) | 11 (4.3) |

* Scoring was reversed for these items

**Table 4. Perceived COVID-19 vaccination effectiveness among HCWs in Ayawaso West Municipality of Greater Accra, Ghana, 2023 (n = 256).**

| Statement | Strongly Disagree (%) | Disagree (%) | Neutral (%) | Agree (%) | Strongly Agree (%) |
|---|---|---|---|---|---|
| Do you think that vaccination against COVID-19 can protect you from contracting COVID-19? | 50 (19.5) | 61 (23.8) | 38 (14.8) | 64 (25.0) | 43 (16.8) |
| Do you think that COVID-19 vaccination can prevent people from getting infected due to herd immunity? | 36 (14.1) | 52 (20.3) | 49 (19.1) | 89 (34.8) | 30 (11.7) |
| Do you think mass vaccination against COVID-19 is justified? | 14 (5.5) | 22 (8.6) | 72 (28.1) | 87 (34.0) | 61 (23.8) |

participants, 188 (73.4%) had a positive perception of COVID-19 vaccine effectiveness whereas 68 (26.6%) had a negative perception.

## Health belief model constructs

Table 5 shows the responses of participants to questions relating to the HBM constructs. The majority of participants scored above the calculated mean scores on all five domains of the HBM. More than three-quarters 226 (88.3%) of participants had a high perception of susceptibility to COVID-19 infection. The Cronbach alpha for the perceived susceptibility scale was 0.82. Again, the majority 208 (81.3%) of participants had a high perception of the severity and seriousness of COVID-19 infection. The calculated Cronbach alpha for the perceived severity and seriousness scale was 0.81. The majority 228 (89.1%) of participants had a high perception of COVID-19 vaccination benefits. The Cronbach alpha for the perceived vaccine benefit scale was 0.91. Additionally, 239 (93.4%) of participants had a high perception of barriers to vaccination. The Cronbach alpha for perceived vaccination barriers was 0.83. With regards to cues to action, 203 (79.3%) had high cues to action and the Cronbach alpha for this scale was 0.83.

## Bivariate analysis between socio-demographic characteristics and COVID-19 vaccination uptake among HCWs

To identify candidate confounding socio-demographic variables for inclusion in multivariable logistic regression analysis, a Chi-square test was performed between potential confounders and the outcome variable (uptake of COVID-19 vaccination). However, no significant associations were found between socio-demographic factors and uptake of COVID-19 vaccination. The results of the bivariate analysis are displayed in Table 6.

## Predictors of COVID-19 vaccination uptake

Univariable binary logistic regression models were used to determine the crude odds ratios (COR). Thereafter, variables found statistically significant at $p < 0.2$ were entered simultaneously into the model to determine the adjusted odds ratios (AOR). To ensure there was no multicollinearity between variables, the variance inflation factor (VIF) was checked, and VIF values ranged from 1.1 to 1.7, indicating that multicollinearity was not present among the predictor variables in our regression model [63]. The model was further assessed for goodness of fit using the Hosmer and Lemeshow test. The model was considered fit since it was insignificant at $p < 0.05$ with the Hosmer-Lemeshow test. The model explained 42.7% (Nagelkerke R square) of the variance in the dependent variable and correctly classified 91.8% of the cases.

In the univariable logistic regression model, healthcare workers with good vaccination knowledge (COR = 3.7; 95% CI: 1.7, 8.0), those with a positive attitude towards the COVID-19 vaccine (COR = 10.2; 95% CI: 4.4, 23.4), and positive perception of vaccine effectiveness (COR = 5.9; 95% CI: 2.8, 12.5), those with high perception of susceptibility to COVID-19

**Table 5. HBM constructs of HCWs in Ayawaso West Municipality of Greater Accra, Ghana, 2023 (n = 256).**

| Variable | Strongly Disagree (%) | Disagree (%) | Neutral (%) | Agree (%) | Strongly Agree (%) | Cronbach alpha |
|---|---|---|---|---|---|---|
| **Perceived susceptibility** | | | | | | 0.82 |
| I am susceptible of getting infected due to my occupational exposure. | 10 (3.9) | 24 (9.4) | 22 (8.6) | 81 (31.6) | 119 (46.5) | |
| There is a great chance to get infected by COVID-19 in the next coming months, especially during the cold season. | 12 (4.7) | 28 (10.9) | 88 (34.4) | 95 (37.1) | 33 (12.9) | |
| Healthy people can get COVID-19. | 8 (3.1) | 17 (6.6) | 22 (8.6) | 117 (45.7) | 92 (35.9) | |
| My status as a health worker makes me more susceptible to contract COVID-19. | 16 (6.3) | 31 (12.1) | 80 (31.3) | 82 (32.0) | 47 (18.4) | |
| I believe that I can protect myself against COVID-19 better than other people. | 10 (3.9) | 31 (12.1) | 58 (22.7) | 114 (44.5) | 43 (16.8) | |
| **Perceived severity and seriousness** | | | | | | 0.81 |
| Although for most people, COVID-19 causes mild illness, it makes some people very ill and can be fatal. | 10 (3.9) | 26 (10.2) | 31 (12.1) | 75 (29.3) | 114 (44.5) | |
| I think COVID-19 is more serious than any other Flu like illness | 9 (3.5) | 31 (12.1) | 60 (23.4) | 103 (40.2) | 53 (20.7) | |
| I would be very sick if I get COVID-19 | 14 (5.5 | 46 (18.0) | 94 (36.7) | 82 (32.0) | 20 (7.8) | |
| If I get COVID-19, I might require hospitalization. | 19 (7.4) | 47 (18.4) | 82 (32.0) | 82 (32.0) | 20 (7.8) | |
| If I get COVID-19, I might die. | 52 (20.3) | 59 (23.0) | 60 (23.4) | 56 (21.9) | 29 (11.3) | |
| **Perceived benefits of vaccination** | | | | | | 0.91 |
| Vaccination is a good idea because it makes me feel less worried about catching COVID-19. | 12 (4.7) | 29 (11.3) | 46 (18.0) | 132 (51.6) | 37 (14.5) | |
| Vaccination decreases my chance of getting COVID-19 or its complications. | 8 (3.1) | 31 (12.1) | 35 (13.7) | 137 (53.5) | 45 (17.6) | |
| Vaccines are considered between the most tested and safe medical products. | 10 (3.9) | 23 (9.0) | 76 (29.7) | 122 (47.7) | 25 (9.8) | |
| When I get vaccinated, I protect my patients, family, and friends from infection. | 8 (3.1) | 24 (9.4) | 25 (9.8) | 120 (46.9) | 79 (30.9) | |
| When I get vaccinated, the whole community benefits by preventing the spread of COVID-19 | 18 (7.0) | 24 (9.4) | 36 (14.1) | 109 (42.6) | 69 (27.0) | |
| COVID-19 vaccination is an effective way to prevent and control COVID-19. | 8 (3.1) | 20 (7.8) | 39 (15.2) | 114 (44.5) | 75 (29.3) | |
| High vaccination coverage globally is required to stop COVID-19 pandemic. | 9 (3.1) | 32 (12.5) | 39 (15.2) | 101 (39.5) | 75 (29.3) | |
| **Perceived barriers** | | | | | | 0.83 |
| I am concerned that the vaccine is new and has not been used before. | 13 (5.1) | 46 (18.0) | 61 (23.8) | 111 (43.4) | 25 (9.8) | |
| I am concerned about the side effects of COVID-19 vaccine. | 3 (1.2) | 19 (7.4) | 30 (11.7) | 144 (56.3) | 60 (23.4) | |
| I am concerned about the efficacy of COVID-19 vaccine. 0 | 6 (2.3) | 23 (9.0) | 42 (16.4) | 145 (56.6) | 40 (15.6) | |
| I am concerned about the safety of COVID-19 vaccine. | 9 (3.5) | 14 (5.5) | 44 (17.2) | 140 (54.7) | 49 (19.1) | |
| I am concerned about the accessibility of COVID-19. vaccines (geographical distribution of vaccination centers) | 7 (2.7) | 34 (13.3) | 62 (24.2) | 119 (46.5) | 34 (13.3) | |
| I am concerned about the availability of COVID-19 vaccine in limited quantities for limited categories of the population. | 6 (2.3) | 31 (12.1) | 67 (26.2) | 124 (48.4) | 28 (10.9) | |
| I am concerned whether if the COVID-19 vaccine is allowed by my religion. | 62 (24.2) | 88 (34.4) | 47 (18.4) | 44 (17.2) | 15 (5.9) | |
| I am concerned about the reliability of the manufacturer and the source of supply. | 13 (5.1) | 21 (8.2) | 55 (21.5) | 129 (50.4) | 38 (14.8) | |
| I am concerned about Ghana health system, and the strategy of distribution of the vaccines | 9 (3.5) | 21 (8.2) | 69 (27.0) | 115 (44.9) | 42 (16.4) | |

*(Continued)*

**Table 5.** (Continued)

| Variable | Strongly Disagree (%) | Disagree (%) | Neutral (%) | Agree (%) | Strongly Agree (%) | Cronbach alpha |
|---|---|---|---|---|---|---|
| I am concerned about vaccine mode of administration (Injection) | 22 (8.6) | 65 (25.4) | 63 (24.6) | 79 (30.9) | 27 (10.5) | |
| I am concerned about the number of doses of the vaccine that I have to take. | 14 (5.5) | 39 (15.2) | 63 (24.6) | 105 (41.0) | 35 (13.7) | |
| I am concerned about the duration of immunity for the vaccine (how much time I will be protected) | 7 (2.7) | 15 (5.9) | 70 (27.3) | 115 (44.9) | 49 (19.1) | |
| **Cues to action** | | | | | | 0.82 |
| Adequate and reliable information was necessary for me before taking the vaccine. | 4 (1.6) | 23 (9.0) | 19 (7.4) | 138 (53.9) | 72 (28.1) | |
| I took the vaccine because it was recommended by the health facilities. | 21 (8.2) | 46 (18.0) | 27 (10.5) | 113 (44.1) | 49 (19.1) | |
| I took the COVID-19 vaccine because it was recommended by a family member. | 50 (19.5) | 133 (52.0) | 44 (17.2) | 22 (8.6) | 7 (2.7) | |
| I took the COVID-19 vaccine because it was recommended by the health authorities. | 18 (7.0) | 38 (14.8) | 42 (16.4) | 101 (39.5) | 57 (22.3) | |
| I took the COVID-19 vaccine because it was widely recommended by the media. | 37 (14.5) | 102 (39.8) | 45 (17.6) | 53 (20.7) | 19 (7.4) | |
| I took the COVID-19 vaccine after recommendations at my work. | 26 (10.2) | 72 (28.1) | 30 (11.7) | 93 (36.3) | 35 (13.7) | |
| I took the COVID-19 vaccine because it was taken by many in the public. | 44 (17.2) | 128 (50.0) | 45 (17.6) | 28 (10.9) | 11 (4.3) | |
| I took the vaccine because it was a requirement by my employers. | 38 (14.8) | 102 (39.8) | 31 (12.1) | 55 (21.5) | 30 (11.7) | |

infection (COR = 11.0; 95% CI: 4.5, 26.8), and high perception of vaccine benefits (COR = 13.3; 95% CI: 5.1, 34.4) and those with high cues to action (COR = 9.6; 95% CI: 4.4, 20.7) were significantly more likely to have received the COVID-19 vaccination. However, after accounting for the effects of other variables in the adjusted model, only attitude towards vaccination and cues to action were independent predictors of COVID-19 vaccination uptake. Healthcare workers with positive attitudes towards the vaccine were 4.3 times more likely to have received the COVID-19 vaccine compared to those with negative attitudes (AOR = 4.3; 95% CI: 1.4, 13.0), and those with high cues to action were 5.7 times more likely to have been vaccinated against COVID-19 compared to those with low cues to action (AOR = 5.7; 95% CI: 2.2, 14.8) (Table 7).

## Discussion

### Uptake of COVID-19 vaccination among HCWs

This study found an outstanding vaccine uptake among HCWs, with more than three-quarters (85.9%) of participants having received at least one dose of the COVID-19 vaccine. The high COVID-19 vaccination rate among HCWs in Ghana is similar to high rates reported by other researchers, such as a vaccination rate of 72.1% in Zambia [39], 77.0% in China [64], 82.5% in Malawi [65], and 70.5% in Egypt [66]. In contrast to these findings, formative studies conducted before the COVID-19 vaccines were rolled out reviewed that most HCWs had concerns about the vaccines and were not willing to get vaccinated once vaccines were made available. In Ghana, for example, a study conducted before the availability of the COVID-19 vaccines found that only 39.3% of HCWs expressed willingness to receive the vaccine [37]. Similarly, an acceptance of 27.7% among HCWs was reported in the Democratic Republic of Congo [67]. Similar trends have been reported in several other formative studies [36, 68–70]. We attribute this inconsistent finding to the nature in which formative studies were conducted. Most were rapid assessments of the situation and employed non-probability sampling techniques,

**Table 6. Bivariate association between socio-demographic characteristics of health workers in Ayawaso West Municipality and uptake of COVID-19 vaccination.**

| Variable | COVID-19 | vaccination | | |
|---|---|---|---|---|
| | Vaccinated N = 220 (85.9%) n (%) | Not vaccinated N = 36 (14.1%) n (%) | Chi-square (χ2) | P-value |
| **Age group** | | | 1.78 | 0.410 |
| Less than 30 years | 85 (84.2) | 16 (15.8) | | |
| 30 to 35 years | 80 (89.9) | 9 (10.1) | | |
| Above 35 years | 55 (83.3) | 11 (16.7) | | |
| **Sex** | | | 2.15 | 0.143 |
| Male | 76 (81.7) | 17 (18.3) | | |
| Female | 144 (88.3) | 19 (11.7) | | |
| **Profession type** | | | 7.78 | 0.098** |
| Nurses | 72 (93.5) | 5 (6.5) | | |
| Midwives | 37 (82.2) | 8 (17.8) | | |
| Medical doctors | 21 (91.3) | 2 (8.7) | | |
| Allied Health workers [b] | 51 (78.5) | 14 (21.5) | | |
| Hospital administration staff | 39 (84.8) | 7 (15.2) | | |
| **Years of Practice** | | | 0.32 | 0.854 |
| <5 years | 104 (86.0) | 17 (14.0) | | |
| 5–10 years | 69 (87.3) | 10 (12.7) | | |
| >10 years | 47 (83.9) | 9 (16.1) | | |
| **Marital Status** | | | 0.03 | 0.853 |
| Married | 82 (85.4) | 14 (14.6) | | |
| Not married | 138 (86.2) | 22 (13.8) | | |
| **Frontline health worker** | | | | |
| Yes | 130 (89.7) | 15 (10.3) | 3.83 | 0.051 |
| No | 90 (81.1) | 21 (18.9) | | |
| **Previous COVID-19 infection** | | | 0.86 | 0.353 |
| Yes | 85 (88.5) | 11 (11.5) | | |
| No | 135 (84.4) | 25 (15.6) | | |

**Fishers Exact test conducted

[b] Includes pharmacists, physiotherapists, dentists, lab technicians, opticians, radiographers, dieticians.

justifiably due to lockdowns, with a high possibility of selection bias. Moreover, our study was conducted during the implementation phase of the vaccination when most HCWs had already been exposed to accurate messages about the vaccines. This exposure to accurate information may have positively influenced their earlier vaccination decisions. Additionally, the influence of peers and colleagues who had received the vaccines within the healthcare community could have played a role in shaping individual choices.

## Knowledge about and attitude toward COVID-19 vaccination among HCWs

HCWs were highly knowledgeable about COVID-19 vaccination. These results are consistent with findings from similar prior studies such as in Nigeria, where health workers were found to be highly knowledgeable about COVID-19 vaccines [34], in Ethiopia where the majority of health workers had more than average knowledge about the vaccines [71] and in Sierra Leone [59]. While our study revealed high overall knowledge among participants regarding COVID-19 vaccination, specific gap in knowledge regarding the eligibility of pregnant and lactating mothers for COVID-19 vaccination existed, indicating the need to implement targeted

**Table 7. Univariable and multivariable logistic regression for factors associated with uptake of COVID-19 vaccination among health workers in Ayawaso West Municipality.**

| Variable | COR[a] (95% CI) | p-value | AOR[b] (95% CI) | p-value |
|---|---|---|---|---|
| **Knowledge about COVID-19 Vaccination** | | **0.001**** | | 0.115 |
| Poor knowledge | 1 | | 1 | |
| Good knowledge | 3.7 (1.7–8.0) | | 2.1 (0.8–5.6) | |
| **Attitude towards COVID-19 vaccination** | | **<0.001**** | | **<0.01*** |
| Negative attitude | 1 | | 1 | |
| Positive attitude | 10.2 (4.4–23.4) | | 4.3 (1.4–13.0) | |
| **Perceived COVID-19 vaccine effectiveness** | | **<0.001**** | | 0.299 |
| Low perception | 1 | | 1 | |
| High Perception | 5.9 (2.8–12.5) | | 1.8 (0.6–5.1) | |
| **Perceived susceptibility to COVID-19** | | **<0.001**** | | 0.165 |
| Low perception | 1 | | 1 | |
| High perception | 11.0 (4.5–26.8) | | 2.6 (0.7–10.4) | |
| **Perceived severity and seriousness of COVID-19** | | 0.138 | | 0.567 |
| Low perception | 1 | | 1 | |
| High perception | 1.9 (0.8–4.4) | | 0.7 (0.2–2.6) | |
| **Perceived COVID-19 vaccine benefits** | 1 | **<0.001**** | 1 | 0.245 |
| Low Perception | 1 | | 1 | |
| High Perception | 13.3 (5.1–34.4) | | 2.5 (0.5–11.2) | |
| **Perceived barriers to COVID-19 Vaccination** | | 0.335 | | |
| High Perception | 1 | | | |
| Low Perception | 2.7 (0.4–21.4) | | | |
| **Cues to action** | | **<0.001**** | | **<0.001*** |
| Low cues | 1 | | 1 | |
| High cues | 9.6 (4.4–20.7) | | 5.7 (2.2–14.8) | |

COR-Crude Odds Ratio, AOR-Adjusted Odds Ratio

*-Statistically significant at p<0.01

**-Statistically significant at p<0.001

[a] -Univariable logistic regression analysis

[b] -Multivariable logistic regression analysis

educational campaigns aimed at improving awareness among healthcare workers regarding the eligibility criteria for pregnant and lactating mothers for COVID-19 vaccination. Most participants had a positive attitude towards COVID-19 vaccination. Similar trends have been reported by Kanu et al among Sierra Leonian health workers [59] and by Tolossa et al among Ethiopian health workers [72]. Contrary to these findings, Alle et al in a study among health workers at a specialized hospital in Ethiopia reported that majority of health workers had an overall negative attitude toward the COVID-19 vaccine [73]. This inconsistency could partly be attributed to the fact that the later study recruited participants from one health facility, hence the results may not be representative of the views of HCWs in Ethiopia.

## Perceived COVID-19 vaccination effectiveness among HCWs

Furthermore, we established that the majority of HCWs perceived the COVID-19 vaccination to be effective in preventing COVID-19 infection. These results were supported by a global survey of twenty countries among the general adult population in which perceived COVID-19 vaccine effectiveness ranged from 67.8% in Egypt to 95.9% in Malaysia [74]. Contrary to our

findings, which conform with findings among the general adult population, a study in Egypt among HCWs found higher levels (67.8%) of disbelief about COVID-19 vaccine effectiveness [75]. This is attributed to the rampant spread of misinformation about the COVID-19 vaccination across the media in Egypt after the vaccine was rolled out, as reported in previous studies [76, 77].

## Determinants of COVID-19 vaccination uptake

Our study established a significant relationship between knowledge about COVID-19 vaccination and uptake of the vaccination in the univariable analysis. Knowledge about vaccination among HCWs has previously been established to be an important determinant of vaccination [40, 41]. With regard to COVID-19, significant associations have been found between knowledge about COVID-19 vaccination and acceptance of the vaccine among health workers in Greece [78], Italy [33], and China [32]. Our study aligns with these findings, as we observed similar trends and patterns, substantiating the conclusions drawn in prior studies. This study further established that attitude towards COVID-19 vaccination was an important determinant of vaccine uptake. Increased positive attitudes towards COVID-19 vaccination independently predicted vaccine uptake. The observed association was documented in prior studies [35, 42, 43, 51].

The HBM has widely been used in determining vaccination behavior, particularly COVID-19 vaccination. Most HBM constructs have significantly been associated with COVID-19 vaccine acceptance [42, 44, 50, 51]. In this current study, five constructs of the HBM provided a framework for assessing determinants of COVID-19 vaccination uptake among HCWs. Those who perceived themselves to be at high risk of getting infected with COVID-19, those who perceived the COVID-19 vaccination to be beneficial, and those with high cues to action were significantly more likely to have received the COVID-19 vaccine in the univariable analysis. Similar findings were reported among HCWs in China [79] and Ethiopia [44] and among the general population in Malawi [80], Israel [46] and Bangladesh [50]. These findings provide an evidence-based formulation of vaccination strategies.

In this study, several sociodemographic factors that have previously been associated with COVID-19 vaccine acceptability were considered, such as age [81], Sex [82] profession type [70], marital status [83], and working in the frontline [58]. However, no significant associations between sociodemographic factors and uptake of the COVID-19 vaccination were found. Similar insignificant associations were reported among Israelites [46]. The lack of significant associations between sociodemographic factors and COVID-19 vaccination uptake in our study may be attributed to several factors. Firstly, the timing and context of the study may have influenced the observed findings. Sociodemographic factors associated with vaccine acceptability can vary based on the stage of the vaccination rollout and prevailing public health measures. While previous studies were conducted before vaccine availability, our study was conducted during the vaccine rollout period when messages regarding COVID-19 vaccination in Ghana were rampant [84, 85]. This widespread dissemination of information may have contributed to increased awareness and acceptance of vaccination across various sociodemographic groups, potentially minimizing the impact of individual characteristics on vaccine uptake.

## Factors independently associated with COVID-19 vaccination uptake

After adjusting for the effects of other variables, cues to action and attitude toward COVID-19 vaccination were the only significant predictors of COVID-19 vaccine uptake. Cues to action and attitude towards COVID-19 vaccination had a direct influence on vaccine uptake, while

knowledge about COVID-19 vaccination, perceived susceptibility, perceived vaccine effectiveness, and benefit might have influenced vaccine uptake through attitude towards vaccination and cues to action. Previous studies established that cues to action have the most significant effect on vaccine acceptance, which makes our observation not surprising. In China, HCWs with high cues to action were 23 times (AOR = 23.66, 95% CI: 9.97–56.23) more likely to accept the COVID-19 vaccine compared to those with low cues to action [86]. Similar conforming results were reported by other researchers in China [79] and Malawi [80]. Cues to action work as "triggers" that prompt individuals to change their behavior [83]. In the context of COVID-19 vaccination, governments, and experts' provision of accurate information about the vaccine acts as triggers to encourage active participation in vaccination initiatives [87]. Ghana employed a combination of informative, motivational, and coercive strategies to address vaccine hesitancy [84]. This strategy could have resulted in high cues to action among HCWs, explaining our findings in this study, and this could have resulted in the higher vaccine uptake reported in this study. This suggests the important role of dissemination of information through government institutions to increase the uptake of COVID-19 vaccination among HCWs.

## Implications for policy and practice

The findings of this study have important implications for policy and practice in the context of COVID-19 vaccination among HCWs in Ghana. Given the complex nature of COVID-19 vaccination influenced by various social, cultural, geographical, and political factors, targeted strategies are essential to address vaccine hesitancy among HCWs. The high vaccine uptake observed among HCWs in Ghana highlights the effectiveness of comprehensive vaccination campaigns and the importance of accurate messaging. To further ensure that all remaining HCWs are vaccinated, the following measures must be put in place.

1. Tailored Vaccination Campaigns: Given the high vaccine uptake observed among HCWs in Ghana, tailored vaccination campaigns should be designed to reinforce the positive attitudes and behaviors identified in this study. Targeted messaging should focus on the effectiveness and benefits of COVID-19 vaccination, particularly highlighting the role of HCWs as frontline protectors against the pandemic.

2. Qualitative Exploration of Barriers: Further qualitative studies are warranted to explore in-depth the barriers to vaccination identified in this study. Understanding the underlying reasons for vaccine hesitancy among HCWs can inform the development of targeted interventions to address specific concerns and misconceptions.

3. Enhance positive attitude about COVID-19 vaccination: It is recommended that targeted interventions be developed to address and enhance positive attitudes towards COVID-19 vaccination. This may involve implementing educational campaigns, training sessions, and interactive workshops aimed at dispelling myths and addressing concerns. Collaborative efforts involving healthcare authorities, professional associations, and media outlets are vital in ensuring the widespread dissemination of accurate information regarding COVID-19 vaccination among HCWs.

4. Leveraging Cues to Action: Strategies should be developed to leverage cues to action, such as peer influence to promote vaccine uptake among HCWs. This could include peer-led vaccination initiatives, educational workshops, and communication campaigns led by trusted healthcare leaders.

### Limitations of the study

We recognize a few limitations to this study. Our study relied heavily on hospital records to recruit a representative sample of HCWs. While this may be considered a strength, we recognize that not all HCWs were documented in the hospital registers, which meant that those omitted from registers were automatically excluded from the study. This may have led to undercoverage. Another limitation lies in the engagement of healthcare workers as data collectors and self-reported vaccination status among HCWs, with a high risk of social desirability bias. We, however, minimized this risk by assuring our participants of confidentiality and anonymizing all questionnaires. Further, we took measures to minimize this bias by ensuring that completed questionnaires were submitted in sealed envelopes, preventing data collectors from viewing participants' responses. Moreover, our scale for measuring knowledge about COVID-19 vaccination had a low internal consistency, with a Cronbach's alpha of 0.52, which is lower than the satisfactory criteria of $\geq 0.7$ [60]. This raises concerns about the reliability of our knowledge scale. We therefore interpret this scale with caution. Despite these limitations, we still believe that our study provides valuable insights that are crucial for promoting COVID-19 vaccination among HCWs in Ghana.

### Conclusion

COVID-19 vaccination among health workers in Ghana is promising. We found a high vaccination rate among HCWs in Ghana. Attitude towards vaccination and cues to action were the two most important factors affecting the uptake of the vaccination. Despite the high uptake of the COVID-19 vaccination among HCWs in Ghana, there is a significant proportion that is still hesitant. To ensure that all HCWs are vaccinated, Interventions to promote vaccination should target key determinants of vaccination uptake, such as attitude towards the vaccination and cues to action.

### Supporting information

**S1 File. Questionnaire.**
(PDF)

### Acknowledgments

We would like to thank healthcare workers who participated in this study. Additionally, we are grateful to Ayawaso West municipal directorate of health for accepting our request to collect data in the district.

### Author Contributions

**Conceptualization:** Whiteson Mbele, Phyllis Dako-Gyeke, Andreas Ndapewa Frans.

**Formal analysis:** Whiteson Mbele.

**Investigation:** Whiteson Mbele.

**Methodology:** Whiteson Mbele, Phyllis Dako-Gyeke, Andreas Ndapewa Frans.

**Supervision:** Phyllis Dako-Gyeke.

**Visualization:** Phyllis Dako-Gyeke.

**Writing – original draft:** Whiteson Mbele, Andreas Ndapewa Frans.

**Writing – review & editing:** Phyllis Dako-Gyeke.

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
