## [Decision Letter · Decision Letter 0]

14 Feb 2024

PGPH-D-23-02363

COVID-19 vaccination uptake among healthcare workers in Ghana: A comprehensive analysis of knowledge, attitude, perceived vaccine effectiveness, and health belief model constructs

Dear Dr. Mbele,

Thank you for submitting your manuscript to PLOS Global Public Health. After careful consideration, we feel that it has merit but does not fully meet PLOS Global Public Health’s publication criteria as it currently stands. Therefore, we invite you to submit a revised version of the manuscript that addresses the points raised during the review process. The manuscript has been evaluated by 5 reviewers, and their comments are available below.

The reviewers have raised a few concerns. They feel the manuscript would benefit from a more thorough rationale for the study outcomes tested in the study and methodological details such as sample sizes, inclusion criteria and statistical analysis methods and a more thorough discussion of the results and their potential implications. Could you please carefully revise the manuscript to address all comments raised?

We look forward to receiving your revised manuscript.

Kind regards,

Annesha Sil, PhD

PLOS Staff Editor

Journal Requirements:

2. We ask that a manuscript source file is provided at Revision. Please upload your manuscript file as a .doc, .docx, .rtf or .tex.

Additional Editor Comments (if provided):

Reviewers' comments:

Reviewer's Responses to Questions

**Comments to the Author**

1. Does this manuscript meet PLOS Global Public Health’s publication criteria? Is the manuscript technically sound, and do the data support the conclusions? The manuscript must describe methodologically and ethically rigorous research with conclusions that are appropriately drawn based on the data presented.

Reviewer #1: Yes

Reviewer #2: Yes

Reviewer #3: Yes

Reviewer #4: Yes

Reviewer #5: Yes

2. Has the statistical analysis been performed appropriately and rigorously?

Reviewer #1: Yes

Reviewer #2: Yes

Reviewer #3: Yes

Reviewer #4: Yes

Reviewer #5: Yes

3. Have the authors made all data underlying the findings in their manuscript fully available (please refer to the Data Availability Statement at the start of the manuscript PDF file)?

Reviewer #1: Yes

Reviewer #2: Yes

Reviewer #3: Yes

Reviewer #4: Yes

Reviewer #5: Yes

4. Is the manuscript presented in an intelligible fashion and written in standard English?

Reviewer #1: Yes

Reviewer #2: Yes

Reviewer #3: Yes

Reviewer #4: Yes

Reviewer #5: Yes

5. Review Comments to the Author

Reviewer #1: The manuscript titled "COVID-19 Uptake among healthcare workers in Ghana: A comprehensive analysis of knowledge, attitude, perceived vaccine effectiveness, and health belief model constructs" is a novel study with significantly distinct contribution to the scientific knowledge. The manuscript worths publishing because it would inform the Ghanaian public health decision on improving COVID-19 through evidence-based research conducted in this kind of study. I therefore recommend the manuscript for publication such that not only the country's would utilize its outcome to improve her COVID-19 vaccine uptake but also to share the experience with other countries to leverage upon for trial. Except for minor typographical errors and incomplete sentence observed, the manuscript is valid for PLOS Global Public Health acceptance for publication.

Reviewer #2: Abstract

Abstract background looks lengthy; authors could summarize.

Line 16 “Consequently, this study assessed uptake of COVID-19 vaccination and associated factors among healthcare workers in Ghana in the post-vaccine roll-out period.”.

What do the authors refer to as “this”?

Line 19, healthcare workers were randomly selected? What type of random sampling approach was used?

Line 20-21 “Bivariable and Multivariable logistic regression was performed using IBM SPSS version 22 to identify predictors of vaccine uptake and a statistical significance was declared at p<0.05”. I suggest the authors change the verb “was” to were. Bivariable and multivariable tests are different. I suggest the sentence is revised. As it reads, it seems as if bivariable analysis can predict the study outcome.

The results seem not to reflect the entirety of the study title. Where are the findings on perceived vaccine effectiveness?

The methods section of the abstract can be written in detail.

Introduction

The introductory section of the paper lacks information about COVID-19 in Ghana. It will be better if COVID information in Ghana is provided (e.g., case detection and strategies for prevention).

Which reference style was used? Lines 73–64 seem to have a mixed referencing style. Authors should check and correct.

The introduction also lacks information on knowledge, attitudes, and perceived effectiveness. The emphasis is laid on vaccine uptake. It is important that readers have a good understanding of the study outcomes in the introduction.

Line 65: Authors can furnish readers with current information; the May 2020 cases on COVID look outdated for a current paper. Authors can update.

Methodology

Line 168–170: The exclusion criteria are not clear. What explicitly defined participant exclusion in the study? Sick, disagreed to participate?

Line 186: How many health workers were from each health facility? This will help readers understand the proportionate allocation. Why only 14 in the 5 CHPS zones?

Lines 187–188: “Secondly, stratification was done by profession type, and the sample size to be recruited from each profession was proportionally allocated.”. Stratification by profession type: how many groups were obtained in the stratification? The sampling approach needs major revision. Which R command was used in the random selection? Line 189: This statement could come earlier than line 185. I asked how you got to know the number of health workers in each facility without first contacting the Municipal health directorate or the human resource department of the facilities.

Authors should be consistent with the use bivariable or bivariate.

Line 242-244 “Bivariate analysis was performed between socio-demographic characteristics and the outcome variable (uptake of COVID-19 vaccination) using the Chi-square test to determine potential confounding variables to include in logistic regression.”. I do not think chi-square test help in determining potential confounders, as stated, it tests an association between variables.

Line 249: What were the VIF values obtained to help readers know multicollinearity never existed in the predictor variables?

Line 224: were the HCW who collected the data from the facilities. A brief description of them will be good.

Discussion

I do not think lines 411–418 are very relevant. The authors can clearly discuss the study findings.

What is the major difference between the heading in line 458 and the heading in line 496? This could be merged. Again this could come before the multivariate discussion.

What could account for the disparity in association of the demographic factors and COVID-19 uptake in the comparison made with previous study (line 497-501). Authors could further explain.

Line 479: Adjusting for in adjusted model seems repetitive, adjusted model considers adjustment, hence no need to repeat.

484–487: Repetition of Reference. The authors have referred and provided elaboration. This could be revised with references 80 and 83.

Limitations

The health workers used in the data collection could have biased responses.

Reviewer #3: It is true that author's findings provide data to an evidence-based formulation of vaccination strategies. Then, concluding that interventions to promote vaccination should target key determinants of vaccination uptake, such as attitude towards the vaccination and cues to action is reasonable. However, it would be great to propose a strategy for that o cite some reference.

Reviewer #4: Introduction

1. The investigators have pointed out infection rates among doctors in Ghana. What was the infection rate among the other cadres of healthcare workers in Ghana. Was it higher than the regional and/or global infection rates? What was the mortality figures for healthcare workers in Ghana? How did it compare with the African or Global figures?

2. You have mentioned inadequate data on the actual vaccination uptake among HCWs post-rollout. What do you mean by inadequate? Was there not even a single study about the actual hesitancy after the roll out? What was the vaccination acceptance rates reported in studies that you accessed? How does these figures compare with the region and global statistics?

3. Although the authors identified the hesitancy among HCWs, they didn’t delve deeper into the reasons behind this hesitancy. Therefore, exploring the factors contributing to vaccine hesitancy among HCWs could enrich the discussion.

4. The conceptual framework is not provided and should be included.

Materials and methods

1. The duration of data collection has been mentioned but it might be beneficial to provide reasoning for the specific timeframe chosen, especially concerning any potential fluctuations or significant events that could affect participants' perceptions or behaviors during the period mentioned.

2. Some of the inclusion and exclusion criteria pointed out are not well thought and should be rewritten or moved to their right section. For instance, consenting to a study should be moved to ethical consideration section.

3. The definition of COVID-19 vaccine uptake is clear; however, specifying the types of vaccines considered might add clarity to the outcome measure, given the availability of different vaccines at different times.

4. The section briefly describes the statistical analysis plan, including bivariate and multivariable logistic regression. Further details on the specific variables included in the model, handling of missing data, and assumptions made could enhance the rigor of the analysis explanation.

5. What informed the choice of statistical significance for univariable regression analysis which was set at p<0.2?

6. The study adequately addresses ethical considerations, including approvals and participant consent. However, detailing steps taken to ensure data confidentiality and participant privacy during data storage and analysis would further strengthen the ethical assurance.

7. The authors should consider adding recall bias or social desirability bias in self-reported responses as part of the limitations.

Results

1. Despite high overall awareness, specific gaps exist, particularly regarding the eligibility of pregnant and lactating mothers for COVID-19 vaccination. How did you handle these parameters?

2. While the study identified various barriers to vaccination, deeper exploration of their impact on vaccination behavior could enhance the practical implications of the findings.

Discussion

1. What are the implications of your findings?

Reviewer #5: The work is exciting and effectively reflects the determinants of COVID-19 vaccine decision-making in a pandemic context. The fact that data collection took place in 2023, unfortunately, directly impacts the results regarding vaccine adherence and knowledge about vaccines. However, the study is well-designed, the sample well-calculated, the randomness of the involved workers well thought out, the questionnaire content well-planned, and the statistics well-executed and presented that the work is delightful to read and ultimately provides powerful insights on the subject. I believe the work has great potential for acceptance, as it aligns with the journal's scope, the language is clear and concise, and the content is relevant.

There are a few points I would address, but they are details. In the introduction, the data regarding the number of cases and deaths is from early 2023. Considering the publication is now, I suggest updating it with the latest data from 2024 or simply updating the data until the data collection in August 2023 (to provide contextualized information).

In line 57, I believe a semicolon should not be there; I suggest leaving the sentence with just commas.

In line 156, the word "healthcare"

---

## [Decision Letter · Decision Letter 1]

20 Mar 2024

COVID-19 vaccination uptake among healthcare workers in Ghana: A comprehensive analysis of knowledge, attitude, perceived vaccine effectiveness, and health belief model constructs

PGPH-D-23-02363R1

Dear Dr Mbele,

We are pleased to inform you that your manuscript 'COVID-19 vaccination uptake among healthcare workers in Ghana: A comprehensive analysis of knowledge, attitude, perceived vaccine effectiveness, and health belief model constructs' has been provisionally accepted for publication in PLOS Global Public Health.

Best regards,

Julia Robinson

Staff Editor

Reviewer Comments (if any, and for reference):

Reviewer's Responses to Questions

**Comments to the Author**

1. If the authors have adequately addressed your comments raised in a previous round of review and you feel that this manuscript is now acceptable for publication, you may indicate that here to bypass the “Comments to the Author” section, enter your conflict of interest statement in the “Confidential to Editor” section, and submit your "Accept" recommendation.

Reviewer #1: All comments have been addressed

Reviewer #2: All comments have been addressed

Reviewer #3: All comments have been addressed

Reviewer #5: All comments have been addressed

2. Does this manuscript meet PLOS Global Public Health’s publication criteria? Is the manuscript technically sound, and do the data support the conclusions? The manuscript must describe methodologically and ethically rigorous research with conclusions that are appropriately drawn based on the data presented.

Reviewer #1: Yes

Reviewer #2: Yes

Reviewer #3: Yes

Reviewer #5: Yes

3. Has the statistical analysis been performed appropriately and rigorously?

Reviewer #1: Yes

Reviewer #2: Yes

Reviewer #3: Yes

Reviewer #5: Yes

4. Have the authors made all data underlying the findings in their manuscript fully available (please refer to the Data Availability Statement at the start of the manuscript PDF file)?

Reviewer #1: Yes

Reviewer #2: Yes

Reviewer #3: Yes

Reviewer #5: Yes

5. Is the manuscript presented in an intelligible fashion and written in standard English?

Reviewer #1: Yes

Reviewer #2: Yes

Reviewer #3: Yes

Reviewer #5: Yes

6. Review Comments to the Author

Reviewer #1: The Author has responded all the comments and corrections specifically those related to the precision and methodology of the study. I would like to recommend the acceptance of the manuscript.

Reviewer #2: (No Response)

Reviewer #3: I consider all comments have been addressed.

Reviewer #5: All my comments have been adressed, congratulations for the work.

7. PLOS authors have the option to publish the peer review history of their article (what does this mean?). If published, this will include your full peer review and any attached files.

**Do you want your identity to be public for this peer review?** For information about this choice, including consent withdrawal, please see our Privacy Policy.

Reviewer #1: **Yes: **Sikiru Olanrewaju Badaru

Reviewer #2: No

Reviewer #3: **Yes: **Jorge Alberto Álvarez Díaz

Reviewer #5: **Yes: **Gabriel Berg de Almeida
